# Prompt-Guided Alignment with Information Bottleneck Makes Image Compression Also a Restorer

**Xuelin Shen**[2]* **Quan Liu**[2,3]* **Jiayin Xu**[2,3] **Wenhan Yang**[1] [†]

[1]Peng Cheng Laboratory
[2]Guangdong Laboratory of Artificial Intelligence and Digital Economy (SZ)
[3]College of Computer Science and Software Engineering, Shenzhen University
`shenxuelin@gml.ac.cn, quanliu@gml.ac.cn`
`jiayinxu@gml.ac.cn, yangwh@pcl.ac.cn`

## Abstract

Learned Image Compression (LIC) models face critical challenges in real-world scenarios due to various environmental degradations, such as fog and rain. Due to the distribution mismatch between degraded inputs and clean training data, well-trained LIC models suffer from reduced compression efficiency, while retraining dedicated models for diverse degradation types is costly and impractical. Our method addresses the above issue by leveraging prompt learning under the information bottleneck principle, enabling compact extraction of shared components between degraded and clean images for improved latent alignment and compression efficiency. In detail, we propose an Information Bottleneck-constrained Latent Representation Unifying (IB-LRU) scheme, in which a Probabilistic Prompt Generator (PPG) is deployed to simultaneously capture the distribution of different degradations. Such a design dynamically guides the latent-representation process at the encoder through a gated modulation process. Moreover, to promote the degradation distribution capture process, the probabilistic prompt learning is guided by the Information Bottleneck (IB) principle. That is, IB constrains the information encoded in the prompt to focus solely on degradation characteristics while avoiding the inclusion of redundant image contextual information. We apply our IB-LRU method to a variety of state-of-the-art LIC backbones, and extensive experiments under various degradation scenarios demonstrate the effectiveness of our design. Code is available at `https://github.com/liuquan0521-sys/IB-LRU-compression`.

## 1 Introduction

In the past decade, Learned Image Compression (LIC) has emerged as a competitive alternative to conventional image coding standards [1, 2, 3] by leveraging deep neural networks to optimize the rate-distortion trade-off in an end-to-end manner. Existing LIC models typically adopt an autoencoder structure, where the encoder extracts a latent representation and the decoder reconstructs the image, characterized by an integrated entropy model that captures spatial dependencies in the latent representation and plays a critical role in compression efficiency. A milestone in entropy modeling is the introduction of the hyperprior [4], which adopts the variational autoencoder (VAE) framework and introduces a Gaussian-based prior to model the distribution of latent representations. This approach enables the compression model to adaptively learn compact latent representations tailored

---

*These authors contributed equally.
[†]Correspondence to: Wenhan Yang.

39th Conference on Neural Information Processing Systems (NeurIPS 2025).

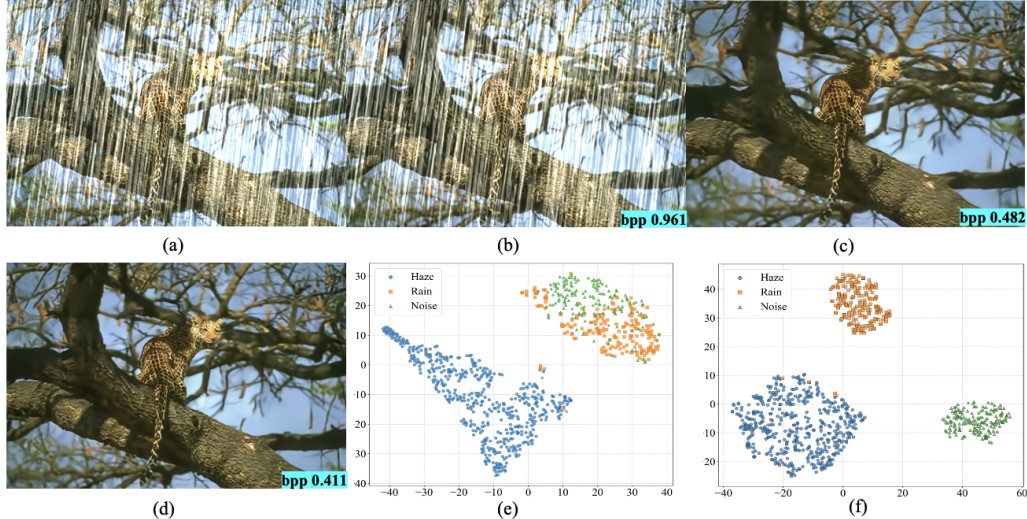

Figure 1: An intuitive illustration of compression performance under degradation scenarios. (a) Input image with rain degradation; (b) Output from a well-trained LIC network [5]; (c) Output from the *restoration-then-compression* paradigm (PromptIR [6] + LIC); (d) Output from our proposed IB-LRU scheme. In addition, we compare the commonly employed factorized prompt [6] (e) and our proposed probabilistic prompt (f) in terms of their capacity to capture discriminative degradation characteristics, visualized via t-SNE plots.

to the statistical properties of natural images, significantly improving rate-distortion performance and forming the foundation for the following LIC models.

Despite their impressive performance under standard conditions, existing LIC models often struggle in real-world settings, where the imaging process typically suffers from various environmental degradations, *e.g.*, noise, fog, rain, or low-light conditions. This limitation arises from their data-driven nature, that are trained on high-quality natural images and relies on learned priors that assume clean image statistics. However, in degraded scenarios, the obtained latent representations deviate significantly from these assumptions. Consequently, as the entropy model is tightly coupled with the prior, it becomes difficult to accurately estimate the distribution of latent features, leading to a substantial drop in compression performance. In other words, compressing degraded images often requires significantly more bits than pristine ones, an intuitive demonstration is provided in Fig. 1 (b). Moreover, the wide range of possible degradation types and intensities in real-world settings makes it impractical to retrain dedicated LIC models for each case to capture their specific distributions.

In facing this challenge, a potential path is *restoration-then-compress* paradigm, which involves deploying cutting-edge *multy-in-one* image restoration models [7, 8, 9, 10] (denoting their capacity to handle multiple degradation within a unified process) at encoding end, with the aim of reducing the degradation in a preprocessing stage so that the restored images can be compressed with an acceptable rate cost. However, in our coding practice, this paradigm does not yield optimal performance. Although it may provide satisfactory perceptual quality, the restored images still exhibit distributional divergence from natural images, leading to suboptimal compression results in the context of LIC, as shown in Fig. 1 (c). Moreover, adding a separate restoration model may reduce flexibility in real-world deployments, especially when computational resources at the imaging end are limited.

To tackle the aforementioned challenges, we introduce the Compressor-as-Restoration paradigm, where prompts are employed to adaptively steer the encoder, thereby achieving a unified framework for compression and restoration. We propose a novel Information-Bottleneck-Constrained Latent Representation Unifying (IB-LRU) scheme, a lightweight encoder-side plugin module that constrains latent representations by aligning degraded and clean-image distributions, thereby enhancing compression and reconstruction without modifying the LIC backbone. In particular, to address diverse input degradations within a unified training framework, we adopt a prompt learning strategy enhanced by a VAE-based Probabilistic Prompt Generator (PPG), which models each degradation using distribution parameters. Unlike existing approaches that rely on factorized prompt vectors, our probabilistic design yields more compact and discriminative representations with fewer parameters,

as shown in Fig. 1 (e) and (f). During inference, degradation information is sampled from the PPG's posterior and passed to a Degradation-Adaptive Gating Modulation (DAGM) module, which guides the encoding process. Additionally, we incorporate the Information Bottleneck (IB) principle to constrain the mutual information between prompts and image content, encouraging the prompt to focus on essential degradation characteristics. In experiments, we implement our IB-LRU scheme on multiple LIC backbones, and extensive results demonstrate that the proposed scheme effectively improves compression performance under various degradation settings.

Our key contributions are as follows,

- To the best of our knowledge, the proposed IB-LRU is the first exploration aimed at improving the efficacy of LIC models under various degradation settings through a lightweight plug-and-play design.

- We propose a novel probabilistic prompt learning strategy that characterizes each degradation type using a set of distribution parameters, rather than factorized vectors, resulting in more discriminative degradation representations.

- We introduce an Information Bottleneck (IB)-based optimization criterion for the probabilistic prompt learning process, and derive a variational approximation bound in theory that guides the design of our optimization strategy.

## 2 Related Work

**Learned Image Compression**. The past decade has witnessed the rapid advancement of Learned Image Compression (LIC), driven by the progress of deep learning technologies. The first attempt was made by Ballé *et al.* [4], introduced an end-to-end autoencoder pipeline, with a factorized entropy model responsible for constraining the compactness of the latent representation. Furthermore, Ballé *et al.* [11] made a significant advancement by introducing a hyperprior into the entropy coding process, enabling the learning of a hierarchical entropy model in which the distribution of latent features is conditioned on a hyper-latent variable. In the following works, numerous efforts have been made to further enhance the entropy model by leveraging contextual dependencies, such as local context [12], global context information [13], and channel-wise context [14, 15] resulting in incremental improvements. In addition, some studies focus on reducing the computational complexity of the entropy coding process, by leveraging checkerboard context models [16] or sparse sampling methodology [17]. Besides efforts aimed at optimizing the entropy model, other works focus on improving the backbone networks by incorporating architectures such as residual networks [18], invertible neural networks [19, 20], and Swin-Transformers [21, 22, 23, 24]. They enable a more expressive latent representation process, leading to notable improvements in compression performance.

**Prompt Learning**. The concept of prompt learning originated from the field of natural language processing, involving in inducing pre-trained language models (*e.g.*, BERT [25] or GPT [26]) to generate answers given cloze-style prompts, extracting information useful for downstream tasks. Subsequent research in prompt learning shifted toward automating this process using labeled data, replacing manually designed prompts with learnable ones. For instance, Jiang *et al.*[27] employed text mining and paraphrasing techniques to generate a pool of candidate prompts, from which the optimal ones were selected based on training accuracy. Meanwhile, other studies[28, 29, 30] proposed to factorize prompts into a set of continuous vectors that can be end-to-end optimized with respect to a given objective, a strategy known as continuous prompt learning. In the field of computer vision, CoOp [31] was one of the earliest works to adopt the continuous prompt learning methodology, demonstrating notable improvements in transfer learning performance. Building on this, Zhou *et al.*[32] further applied prompt learning techniques to image classification, achieving significant gains in generalization capability. Similarly, Ju *et al.*[33] explored the use of prompt learning to reformat video-related tasks in alignment with pre-training objectives. Among existing prompt learning works for machine vision, the most relevant to ours is PromptIR [6], which uses a set of factorized vectors as prompts to capture the characteristics of different degradation types for a multi-in-one image restoration network. However, the naive prompt representation and learning strategy yield suboptimal performance in capturing clear and distinct features of various degradation.

**Information Bottleneck** The Information Bottleneck (IB) principle, originally proposed by Tishby *et al.* [34], offers a theoretical framework for extracting the most relevant information from input

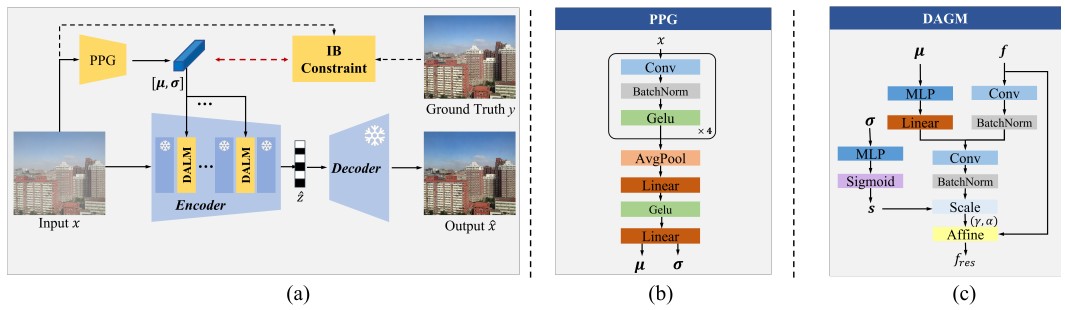

Figure 2: (a) Compression pipeline of the proposed IB-LRU scheme. (b) Details of the proposed PPG. (c) Details of the proposed DAGM module.

data with respect to the task target by compressing the input representation while preserving task-relevant features. In the context of deep learning, the IB framework has been adapted to encourage models to learn compact and generalizable representations by constraining the mutual information between the input and intermediate features[35]. In particular, this work introduced the variational information bottleneck (VIB), which formulates the IB objective in a tractable form using variational approximations, enabling its integration into neural network training pipelines. More recently, the IB principle has been employed in computer vision tasks to enhance robustness and generalization [7, 36, 37, 38]. These works inspire our use of IB to guide prompt learning, aiming to extract clear degradation-relevant information while suppressing redundancy from irrelevant image context.

# 3 Methodology

## 3.1 Motivation and Overview

Existing LIC models typically follow an autoencoder-like pipeline. An encoder $G_a(\cdot)$ maps the input image $x$ to a latent representation $z = G_a(x)$, which is then fed to a uniform quantizer that converts $z$ to its discrete form $\hat{z}$. Subsequently, an entropy encoder $R(\cdot)$ losslessly encodes $\hat{z}$ into a binary bitstream, transmits it to the decoder side, and provides its rate estimation. At the decoder side, the decoder $G_s(\cdot)$ reconstructs the image $\hat{x} = G_s(\hat{z})$. The entire framework is optimized under the *rate-distortion* criterion,

$$\mathcal{L}_{rd} = R(\hat{z}) + \lambda D(x, \hat{x}), \tag{1}$$

where $R(\hat{z})$ is the estimated *rate* measured in bits per pixel (bpp), $D(\cdot, \cdot)$ denotes the *distortion* metric (*e.g.*, MSE: $||x - \hat{x}||^2$), and $\lambda$ is a Lagrange parameter, being responsible for obtaining codecs at different compression levels. However, while LIC models perform well on clean natural images, their effectiveness degrades in real-world conditions involving noise, fog, blur, *et al*. This is because:

- The latent distributions of degraded images deviate significantly from those of clean images;
- The entropy model relies heavily on prior assumptions, making accurate distribution estimation challenging under degradation.

To address the generalization issues of LIC models, we propose an Information-Bottleneck-Constrained Latent Representation Unifying (IB-LRU) scheme, whose central idea is to perform prompt optimization with information bottleneck constraint to guide the encoding of degraded images so that the distribution of their latent representations aligns with that of pristine images, thereby improving compactness and ensuring high-quality outputs at the decoder side. In general, the proposed IB-LRU scheme is designed with three key objectives: flexibility through a lightweight plug-in design; compatibility with existing LIC backbones without requiring parameter modification; and generalizability to support diverse degradation types within a single training process.

The overall framework is illustrated in Fig. 2. The proposed IB-LRU scheme includes two main components: 1) **Probabilistic Prompt Generator** (**PPG**): A VAE-based encoder that learns degradation-specific prompts using a set of distribution parameters drawn from a multivariate Gaussian. At inference time, degradation representations are sampled and used to guide the encoder; 2) **Degradation-Adaptive Gating Modulation** (**DAGM**): A modulation module that adjusts the encoding process using the sampled prompt, effectively aligning the latent distribution across degradation types. More-over, to promote probabilistic prompt learning to capture clear and distinct representations of different

degradations, we investigate: 3) **Information Bottleneck (IB) Principle**: a variational approximation of IB in our context is explored to constrain the mutual information between the learned prompt and the input image while preserving the task-relevant information, thus suppressing redundant information of the learned prompt from the broader image context. The detailed implementations of the proposed PPG and DAGM are presented in Subsection 3.2, while the derivation of the variational approximation of IB is provided in Subsection 3.3.

## 3.2 Prompt-Guide Latent Representation Alignment

**Probabilistic Prompt Generator:** As shown in Fig. 2 (b), the PPG involves a a VAE-style encoder, denoted as $\mathbf{p} = G(x; \boldsymbol{\theta})$, where $\boldsymbol{\theta}$ represents the model parameters and $\mathbf{p}$ is the probabilistic prompt characterized by its distribution parameters $[\mu_{\mathbf{p}}, \sigma_{\mathbf{p}}]$. In particular, during the inference stage, they are sampled from the approximate posterior $q(\mathbf{p}|x; \boldsymbol{\theta}) = \mathcal{N}(\mathbf{p}; \mu_{\mathbf{p}}, \text{diag}(\sigma_{\mathbf{p}}^2))$, where $\mu_{\mathbf{p}} \in \mathbb{R}^d$ is the mean vector, and the diagonal covariance matrix is given by $\text{diag}(\sigma_{\mathbf{p}}^2)$, with $\sigma_{\mathbf{p}}^2 \in \mathbb{R}^d$ representing the per-dimension variances. Herein, $d$ denotes the dimensionality of the prompt.

**Degradation-Adaptive Gating Modulation:** These posterior statistics, $\mu_{\mathbf{p}}$ and $\log \sigma_{\mathbf{p}}^2$, are then fed into a gated modulation process, as shown in Fig. 2 (c). Given an intermediate feature $f$ from a frozen backbone LIC encoder, a modulation network $\text{M}_{\text{mod}}(\cdot)$ tasks as input the $\mu_{\mathbf{p}}$ and $f$ to produce basis modulation maps $[\gamma_{\text{b}}, \alpha_{\text{b}}]$:

$$[\gamma_{\text{b}}, \alpha_{\text{b}}] = \text{M}_{\text{mod}}(\mu_{\mathbf{p}}, f). \tag{2}$$

Concurrently, an MLP layer computes a gating signal $s$ from $\log \sigma_{\mathbf{p}}^2$, which provide information related to the spread of the approximate posterior. This signal $s$ derives final modulation maps $[\gamma, \alpha]$:

$$\begin{aligned} s &= \text{sigmoid}(\text{MLP}(\log \sigma_{\mathbf{p}}^2)), \\ \gamma &= 1 + s \odot (\gamma_{\text{b}} - 1), \\ \alpha &= s \odot \alpha_{\text{b}}. \end{aligned} \tag{3}$$

The restored feature $f_{\text{res}}$ is then obtained via an affine transformation:

$$f_{\text{res}} = \gamma \odot f + \alpha. \tag{4}$$

## 3.3 Information Bottleneck-based Prompt Learning Constraint

As for the training process of the proposed PPG, we posit that an effective prompt $\mathbf{p}$ should act as a minimal sufficient statistic of the degradation signal relative to the content already captured in the intermediate feature $f$ of the frozen backbone LIC encoder. Directly learning $\mathbf{p}$ might lead it to capture redundant information, *e.g.*, image content information, which would inevitably prevent the learned prompt from capturing clean and distinct characteristics of different degradation types. Therefore, we investigate the Information Bottleneck (IB) principle [34] to provide reliable and effective guidance for the prompt learning process. In the context of IB, we aim to find a prompt $\mathbf{p}$ that minimizes the mutual information it retains about the input $x$ while maximizing the mutual information relevant to the target task $y$, which denotes the pristine ground truth in this case. We formulate the IB-based optimization objective for $\mathbf{P}$ [3] as:

$$\mathcal{L}_{\text{IB\_prompt}} = \mathcal{I}(\mathbf{P}; X) - \beta \mathcal{I}(\mathbf{P}; Y|F), \tag{5}$$

where the left term represents the information (potentially redundant) about the input $X$ that $\mathbf{P}$ contains, and the right term quantifies the additional information $\mathbf{P}$ offers about the target $Y$ beyond that already captured by the backbone features $F$, $\beta$ denotes the trade-off parameter.

As directly optimizing Eq. (5) is intractable, we follow Variational Information Bottleneck (VIB) [35] to derive tractable variational upper and lower bounds for the left and right terms, respectively.

**Variational Lower Bound for $\mathcal{I}(\mathbf{P}; Y|F)$:** By definition, $\mathcal{I}(\mathbf{P}; Y|F) = \mathcal{H}(Y|F) - \mathcal{H}(Y|\mathbf{P}, F)$. Since $F$ is a deterministic function of $X$ (frozen), $\mathcal{H}(Y|F)$ can be treated as constant with respect to $\mathbf{P}$. Thus, maximizing $I(\mathbf{P}; Y|F)$ is equivalent to minimizing the conditional entropy $\mathcal{H}(Y|\mathbf{P}, F)$:

$$\mathcal{H}(Y|\mathbf{P}, F) = -\mathbb{E}_{(\mathbf{p}, f) \sim p(\mathbf{p}, f)} \mathbb{E}_{y \sim p(y|\mathbf{p}, f)}[\log p(y|\mathbf{p}, f)]. \tag{6}$$

---

[3] In the following derivations, $X$, $Y$, $\mathbf{P}$, $F$ represent random variables, while $x$, $y$, $\mathbf{p}$ and $f$ are scalar or single instances of random variables

We apply a variational approximation $q(y|\mathbf{p}, f)$ for $p(y|\mathbf{p}, f)$, corresponding to our decoder. Using the non-negativity of KL divergence, $D_{\mathrm{KL}}[p(y|\mathbf{p}, f) \parallel q(y|\mathbf{p}, f)] \geq 0$, we have for any given $\mathbf{p}, f$:

$$\mathbb{E}_{p(y|\mathbf{p},f)}[\log p(y|\mathbf{p}, f)] \geq \mathbb{E}_{p(y|\mathbf{p},f)}[\log q(y|\mathbf{p}, f)]. \tag{7}$$

Substituting this inequality (after taking expectation over $p(\mathbf{p}, f)$) into the entropy expression yields an upper bound for the conditional entropy:

$$\mathcal{H}(Y|\mathbf{P}, F) \leq -\mathbb{E}_{p(y,\mathbf{p},f)}[\log q(y|\mathbf{p}, f)]. \tag{8}$$

Therefore, a variational lower bound for the mutual information is:

$$\mathcal{I}(\mathbf{P}; Y|F) \geq \mathbb{E}_{p(y,\mathbf{p},f)}[\log q(y|\mathbf{p}, f)]. \tag{9}$$

Given the above derivation, we found that maximizing this lower bound is equivalent to maximizing the expected log-likelihood $\mathbb{E}_{p(y,\mathbf{p},f)}[\log q(y|\mathbf{p}, f)]$. Thus, in actual implementation, the *distortion* term in Eq. (1) is employed and acting as a proxy for $-\log q(y|\mathbf{p}, f)$.

**Variational Upper Bound for $\mathcal{I}(\mathbf{P}; X)$:** We aim to derive a tractable upper bound for the mutual information $\mathcal{I}(\mathbf{P}; X)$. By definition, we obtain:

$$\begin{aligned}
\mathcal{I}(\mathbf{P}; X) &= \mathbb{E}_{(x,\mathbf{p}) \sim p(x,\mathbf{p})}\left[\log \frac{p(\mathbf{p}|x)}{p(\mathbf{p})}\right] \\
&= \mathbb{E}_{(x,\mathbf{p}) \sim p(x,\mathbf{p})}[\log p(\mathbf{p}|x)] - \mathbb{E}_{\mathbf{p} \sim p(\mathbf{p})}[\log p(\mathbf{p})].
\end{aligned} \tag{10}$$

As the true prior $p(\mathbf{p}) = \mathbb{E}_{x \sim p(x)}[p(\mathbf{p}|x)]$ is intractable, we introduce $r(\mathbf{p})$, a standard Gaussian distribution, as a approximation to the true prior $p(\mathbf{p})$: Using the non-negativity of the KL divergence between the true prior and our variational approximation, $D_{\mathrm{KL}}[p(\mathbf{p}) \parallel r(\mathbf{p})] \geq 0$, we have:

$$-\mathbb{E}_{\mathbf{p} \sim p(\mathbf{p})}[\log p(\mathbf{p})] \leq -\mathbb{E}_{\mathbf{p} \sim p(\mathbf{p})}[\log r(\mathbf{p})]. \tag{11}$$

Substituting this inequality back into Eq. (10):

$$\mathcal{I}(\mathbf{P}; X) \leq \mathbb{E}_{(x,\mathbf{p}) \sim p(x,\mathbf{p})}[\log p(\mathbf{p}|x)] - \mathbb{E}_{\mathbf{p} \sim p(\mathbf{p})}[\log r(\mathbf{p})]. \tag{12}$$

This bound still involves the true posterior $p(\mathbf{p}|x)$ inside the expectations. Now, we further approximate the expectation involving the true posterior by replacing $p(\mathbf{p}|x)$ with our PPG $q(\mathbf{p}|x; \boldsymbol{\theta})$. This step makes the bound computable using samples from the encoder. Recognizing the structure resembles the KL divergence, we arrive at the commonly used VIB upper bound for mutual information:

$$\begin{aligned}
\mathcal{I}(\mathbf{P}; X) &\leq \mathbb{E}_{x \sim p(x)} \mathbb{E}_{\mathbf{p} \sim p(\mathbf{p}|x)}\left[\log \frac{p(\mathbf{p}|x)}{r(\mathbf{p})}\right] \\
&\approx \mathbb{E}_{x \sim p(x)} D_{\mathrm{KL}}[q(\mathbf{p}|x) \parallel r(\mathbf{p})].
\end{aligned} \tag{13}$$

Things have to be mentioned that, although replacing the true posterior $p(\mathbf{p}|x)$ with the approximate posterior $q(\mathbf{p}|x; \boldsymbol{\theta})$ inside the expectation introduces an additional approximation beyond the rigorous bound derived using $D_{\mathrm{KL}}(p \parallel r) \geq 0$, Eq. (13) provides a practical objective for IB implementations.

With the trackable IB-based optimization creation for prompt learning, the final loss function for our IB-LRU scheme is formulated as,

$$\mathcal{L}_{\mathrm{total}} = \mathcal{L}_{\mathrm{rd}} + \beta \, \mathbb{E}_{x \sim p(x)} D_{\mathrm{KL}}[q(\mathbf{p}|x) \parallel r(\mathbf{p})]. \tag{14}$$

## 4   Experiments

In the experimental stage, we first implement the proposed IB-LRU scheme on multiple backbone LIC networks and evaluate its effectiveness across a variety of degradation settings. Subsequently, we compare its performance with the *restoration-then-compression* paradigm, which is built upon cutting-edge *multy-in-one* restoration models. Moreover, comprehensive ablation studies are conducted to validate the effectiveness of our design.

Table 1: Comparison between the proposed scheme and the *restoration-then-compression* paradigm regrading parameters and inference speed.

| Model | AirNet+TIC | Restormer+TIC | PromptIR+TIC | MOCE-IR+TIC | Ours+TIC |
|---|---|---|---|---|---|
| Extra_Param. | 8.93M | 26.12M | 35.59M | 25.35M | 4.6M |
| Inference Speed (ms) | 483.3 | 363.9 | 416.5 | 349.4 | 109.5 |

Figure 3: Performance comparison between the LIC backbone networks with and without the integration of the proposed IB-LRU scheme.

## 4.1 Experimental Setting

**Benchmark:** The performance is assessed across three degradation types, including haze, rain, and noise, with multiple degradation intensities employed for each type: 1) **Rain:** The Rain100 dataset [39] consists of 2,000 degraded-clean pairs for each of the *heavy* and *light* rain scenarios. In our experiments, 1,800 pairs are used for training and 200 pairs for testing in each scenario. 2) **Noise:** For the noise scenarios, the training set was constructed using 400 images from BSD400 [40] and 4,744 images from WED [41], with degraded versions generated by adding Additive White Gaussian Noise (AWGN) with $\sigma \in \{15, 25, 50\}$. The testing set is formed by combining BSD68 [42] and Urban100 [43], using the same noise settings. 3) **Haze:** The SOTS dataset [44] is employed, with 72,135 / 500 images for training /testing. The training sets of all above datasets are combined to support the training of our IB-LRU scheme.

**LIC backbone:** Three widely adopted LIC networks are employed, including the milestone Bmshj2018 [11], as well as two cutting-edge models: the Swin Transformer-based TIC [5] and MLIC++ [45]. These three LIC backbone networks were pre-trained exclusively on the clean COCO 2017 dataset. For each model, four checkpoints corresponding to different compression levels are employed, and their parameters are kept frozen during the training of our IB-LRU scheme.

**Anchors:** Four cutting-edge *multy-in-one* image restoration networks are employed for the *restoration-then-compression* paradigm, including AirNet[8], Restormer[9], and PromptIR[6], and MOCE-IR[46]. All models are trained from scratch using the same training set as ours to ensure fair comparisons. The restored output images are then fed into well-trained LIC backbones for evaluation.

**Evaluation Criteria:** We introduce *rate-perception* criterion for this unique task. In this context, *perception* refers to the distance between decoded images and corresponding pristine ground truths, measured in terms of PSNR (dB), while the bit-per-pixel (bpp) value is used to quantify the *rate*.

**Implementation Details:** We train our model using the AdamW optimizer with settings $\beta_1 = 0.9$ and $\beta_2 = 0.999$. The initial learning rate is set to $1 \times 10^{-4}$. This rate is maintained for the first 80 epochs (80% of the total training) and then decayed to $1 \times 10^{-5}$ for the remaining 20 epochs. The

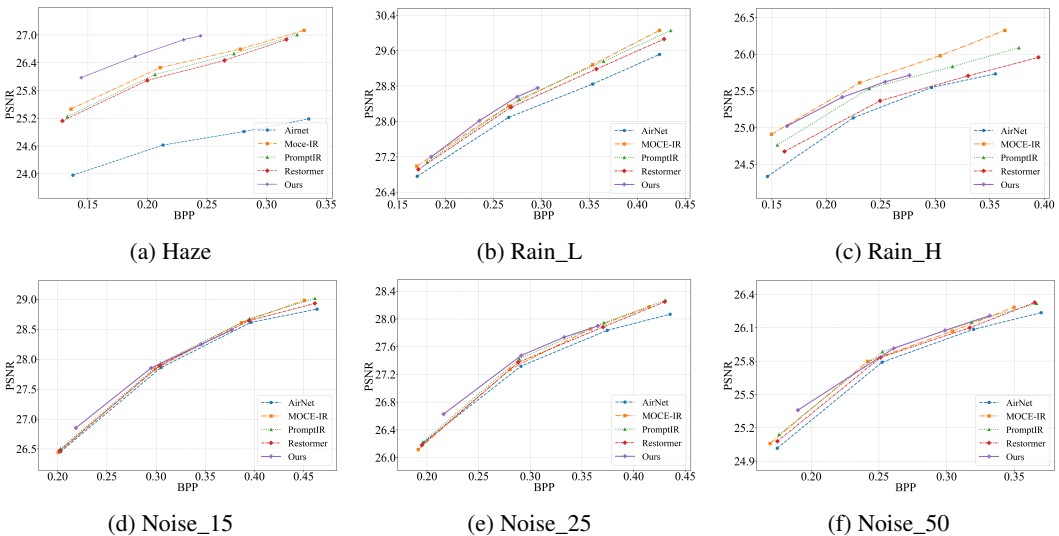

Figure 4: Performance comparison between the *restoration-then-compression* paradigm and our IB-LRU scheme, both of which are based on the TIC backbone.

model is trained for a total of 100 epochs. Training is performed on randomly cropped $256 \times 256$ patches using a setup of 8 NVIDIA RTX 6000 GPUs.

## 4.2 Experimental Results

**Effectiveness Evaluation:** Performance is compared between the LIC backbones with and without our IB-LRU implementation to examine the effectiveness of the proposed scheme. It is worth noting that although the LIC backbones are not designed for image restoration, we focus on bitrate savings in this part, while still reporting their *perception* indices to ensure consistent and clear representation.

The corresponding results, presented in Fig. 3, show encouraging improvements. In particular, under the *light rain*, *heavy rain*, *noise_15*, *noise_25*, and *noise_50* scenarios, our proposed method brings notable improvements in *rate* performance, resulting in average bpp reductions of 27.1%, 51.6%, 18.2%, 34.6%, and 64.9%, respectively, across the three LIC backbones. Moreover, significant improvements in perceptual quality can also be observed, demonstrating the contribution of latent representation unification to the image reconstruction process. It is worth noting that in the *haze* scenario, although the IB-LRU scheme remarkably improves perceptual quality, the bpp values are even increased. According to our analysis and a comprehensive examination of the corresponding dataset, this is primarily because haze tends to act as a smoothing effect on image content rather than introducing additional degradation signals. As a result, hazy images may still be well represented by the prior of the LIC model, as they resemble smooth images.

**Comparison with *Restoration-then-Compression* Paradigm:** Comparisons results between our propose IB-LRU and the *restoration-then-compression* methods regarding *rate-perception* performances are provided in Fig. 4. As shown, our scheme is capable of achieving competitive perceptual quality compared to these cascaded approaches, while offering overall advantages in bpp reduction—especially in high bitrate regions. In particular, an interesting observation arises from the fact that both our IB-LRU and the *restoration-then-compression* approaches are built upon the same frozen LIC checkpoint. While both are capable of achieving satisfactory perceptual quality, they tend to result in a significant increase in bpp. As previously discussed, the underlying reason is that although the restored images may align well with perceptual preferences, their distributions still diverge considerably from those of natural images due to the generative nature of the restoration models. Consequently, during entropy modeling, these distributions cannot be accurately estimated, leading to increased bitrate costs. Meanwhile, our method focuses on unifying the latent distribution, achieving a better trade-off between *rate* and *perception*. To provide intuitive insight into the performance, a set of visual examples is presented in Fig. 5.

To further demonstrate the *flexibility* of our proposed IB-LRU scheme, we compare the number of additional parameters and inference speed with those of the *restoration-then-compression* paradigm.

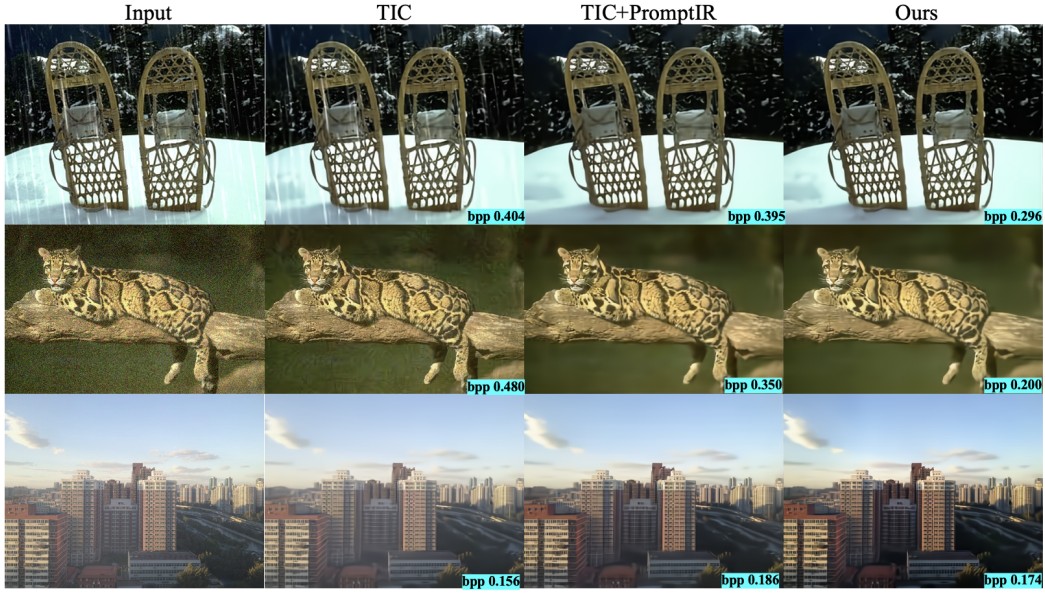

Figure 5: An intuitive illustration of compression performance under different degradation scenarios. The first, second, and third rows correspond to rain, noise, and haze settings, respectively.

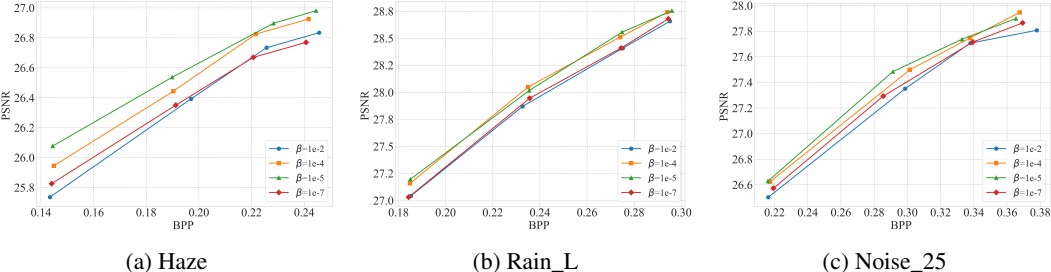

|  (a) Haze | (b) Rain_L | (c) Noise_25 |
|---|---|---|

Figure 6: Ablation results on the IB constraint by varying its weight parameter $\beta$.

The corresponding results, presented in Table 1, demonstrate our overwhelming advantages and highlight the strong potential of IB-LRU for real-world implementation.

## 4.3 Ablation Studies

This part specifically examines the effectiveness of the IB constraint and its influence on the compression pipeline. In particular, the IB principle is employed to regularize the probabilistic prompt, guiding it to capture minimal yet sufficient degradation information. In this part, we retrain the entire scheme with varying values of $\beta$ in Eq. (14) based on the TIC backbone, specifically $\{1 \times 10^{-2}, 1 \times 10^{-4}, 1 \times 10^{-5}, 1 \times 10^{-7}\}$ to examine the influence of the IB constraint. Fig. 6 illustrates the corresponding results, where we observe that the optimal *rate-perception* performance is generally achieved when $\beta$ is set to $1 \times 10^{-4}$ or $1 \times 10^{-5}$, whereas values that are too large or too small lead to suboptimal results.

To gain deeper insight,

- When $\beta$ is too small (*e.g.*, $1 \times 10^{-7}$), the IB constraint becomes weak, allowing the prompt to capture excessive redundant image context rather than distinct and clear degradation characteristics. This, in turn, compromises latent representation unification, as well as compression efficiency and restoration quality.

- Increasing $\beta$ to $1 \times 10^{-5}$ and $1 \times 10^{-4}$ significantly improves *rate-perception* performance, as a moderate IB constraint effectively regularizes the prompt, guiding it to discard irrelevant information while retaining what is essential for adaptive restoration.

- When $\beta$ becomes too large (*e.g.,* $1 \times 10^{-2}$), the *rate-perception* performance begins to degrade. This suggests that overly strong IB constraints may excessively compress the prompt, resulting in loss of useful information necessary for capturing degradation characteristics.

## 5   Conclusion

We propose IB-LRU, a lightweight and modular scheme that enhances the robustness of Learned Image Compression (LIC) models under various real-world degradations. By leveraging a probabilistic prompt guided by the Information Bottleneck principle, our method constrains latent representations to retain task-relevant degradation information while suppressing redundancy. IB-LRU operates without altering the original LIC parameters and is compatible with multiple pre-trained backbones. Empirical results demonstrate consistent gains in compression and restoration performances across diverse degradation scenarios, with minimal additional computational cost. Our findings suggest that latent distribution alignment is a promising direction for improving the generalization of LIC models in practical settings.

## Acknowledgements

This work was in part by the Interdisciplinary Frontier Research Project of PCL under Grant 2025QYB013, in part by the Major Key Project of PCL (PCL2025A03), in part by Guangdong Basic and Applied Basic Research Foundation under Grant 2024A1515010454, in part by the Open Research Fund from Guangdong Laboratory of Artificial Intelligence and Digital Economy (SZ) under Grant No. GML-KF-24-27, in part by the Natural Science Foundation of Guangdong Province under Grant 2023A1515011667.

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

# A   Supplementary Experiments and Analyses

## A.1   Effectiveness of the Probabilistic Prompt Design

This section provides a straightforward comparison between our probabilistic prompt and the existing factorized prompt design. Specifically, we replace our probabilistic prompt with the factorized prompt from PromptIR, keep the feature-modulation module unchanged, and retrain the plug-in on the TIC backbone. Table 2 reports PSNR (dB) at a single compression level, where our methods deliver overall performance gains with significantly fewer parameters.

Table 2: Performance and parameter comparison between our proposed probabilistic prompt and the existing factorized prompt.

| Prompt Type | Rain_L (bpp=0.23) | Haze (bpp=0.19) | Noise_50 (bpp=0.26) | Para. (M) |
|---|---|---|---|---|
| Probabilistic | 28.01 | 26.53 | 25.93 | 4.6 |
| Factorized | 27.95 | 26.39 | 25.76 | 16.0 |

We also assess generalization on unseen conditions, including a mixed haze-and-rain setting created by adding synthetic rain streaks to the SOTS haze test images, and the unseen DID de-raining dataset. Table 3 shows that our method achieves stronger robustness in both cases.

Table 3: Comparison of generalization performance between probabilistic and factorized prompt designs regarding bpp/ PSNR(dB).

| Prompt Type | Haze + Rain (Mixed) | Unseen Domain (DID) |
|---|---|---|
| Probabilistic | 0.183 / 24.09 | 0.202 / 25.15 |
| Factorized | 0.207 / 19.24 | 0.214 / 22.93 |

## A.2   Effectiveness of the Information Bottleneck (IB) Constraint

We evaluate the impact of the Information Bottleneck by removing it entirely ($\beta = 0$). As reported in Table 4, the absence of the IB constraint leads to a uniform degradation in performance, supporting its role in guiding the prompt to capture essential degradation features.

Table 4: Effect of removing the IB constraint ($\beta = 0$) on rate–distortion (bpp / PSNR).

| Version | Rain_L | Haze | Noise_25 |
|---|---|---|---|
| Ours (Full Model) | 0.235 / 28.01 | 0.191 / 26.53 | 0.289 / 27.51 |
| Ablated ($\beta = 0$) | 0.232 / 27.61 | 0.200 / 26.00 | 0.289 / 26.99 |

## A.3   Comparison with Joint Denoising-Compression Methods

We compare against the joint denoising–compression method of Brummer *et al.* [47] (JDC-CN). For a fair comparison, we retrain both approaches on the same TIC backbone using only the *Gaussian noise* split of our training set. As shown in Table 5, our method demonstrates a notable advantage even in this single-degradation setting.

Table 5: Comparison with joint denoising–compression method (bpp/PSNR(dB)).

| Method | Noise_15 | Noise_25 | Noise_50 |
|---|---|---|---|
| Ours | 0.271 / 28.23 | 0.294 / 28.07 | 0.245 / 26.53 |
| JDC-CN [47] | 0.254 / 24.71 | 0.283 / 24.20 | 0.265 / 23.01 |

