# OpenReview forum: "Prompt-Guided Alignment with Information Bottleneck Makes Image Compression Also a Restorer"
_NeurIPS.cc/2025/Conference — NeurIPS 2025 poster_

### Official Review · Reviewer_p8XG · 2025-06-16

**Clarity:** 3
**Significance:** 3
**Originality:** 3
**Rating:** 5
**Confidence:** 4

**Summary:**

The paper presents IB-LRU, a method using IB-constrained prompt learning (via PPG) to dynamically adapt pre-trained image compressors to handle fog, rain, etc., by focusing prompts purely on the degradation, leading to better latent alignment and compression without costly retraining.

**Questions:**

see the Weaknesses

**Ethical Concerns:**

["NO or VERY MINOR ethics concerns only"]

**Final Justification:**

The author has addressed all my concerns, and I am willing to increase the score.

**Limitations:**

yes

**Quality:**

3

**Strengths And Weaknesses:**

Strengths:
(1) The article presents its content clearly and understandably.
(2) When applying LIC models trained on clean image data to degraded image data, their performance significantly deteriorates. However, the alternative approach of first applying image restoration models to degraded images and then compressing the restored images incurs high computational costs (e.g., large parameter count, excessive inference time). Thus, the proposed idea of using a prompt-based scheme to fine-tune pre-trained LIC models is well-justified. Furthermore, empirical results confirm that the proposed scheme achieves superior LIC metrics while maintaining advantages in both parameter count and inference time.
(3) Employing Information Bottleneck theory for prompt design demonstrates significant novelty.

Weaknesses:
(1) A performance comparison between the proposed prompt scheme and existing prompt schemes is not yet provided in the paper.
(2) As shown in Figures 4 and 6, the proposed method outperforms the baselines on some tasks even without the Information Bottleneck loss. Does this suggest that introducing prompts, rather than the IB loss, is the key factor for performance improvement?

---

> ### Author Rebuttal · Authors · 2025-07-31
>
> First of all, we would like to express our sincere appreciation for your responsible and valuable review. Your positive attitude and recognition of our work are a great encouragement to us. We will carefully consider all of your suggestions and do our best to incorporate them into our revised manuscript to further improve its quality.
>
> Q1. Lacking of comparisons with existing prompt schemes.
>
> Ans.: Thank you for your valuable suggestions. A direct comparison with existing factorized prompt schemes is indeed necessary and important. As noted in our manuscript, PromptIR is the most closely related work to ours. Our key contribution lies in further improving prompt design, learning strategy, and the underlying information-theoretic foundation. To this end, we replaced our PPG and DAGM modules with PromptIR’s prompt module and retrained the model under the rate–distortion criterion, using identical hyperparameters for the LIC backbone (TIC). The comparsion performance at the same bpp level and the corresponding plugin module sizes are shown below, where our method shows clear overall improvement.
>
> |PSNR|Rain_L(bpp=0.235)|Haze(bpp=0.191)|Noise_50(bpp=0.261)|Para.(M)|
> |-|-|-|-|-|
> |Probabilistic (Ours)|28.01|26.53|25.93|4.6|
> |Factorized|27.95|26.39|25.76|16.0|
>
> Due to the rebuttal limitations, we are unable to present the full bpp–PSNR curves here. However, we will include comprehensive results and ablation studies in the revised manuscript. We appreciate your understanding.
>
> Q2. Questioning whether the performance gain is mainly due to the prompt design rather than the IB loss.
>
> Ans.: Thanks a lot for you valuable suggestion.  Herein, we would first like to clarify that the ablation studies in Fig. 6 investigate the influence of different β values from 1e-7 to 1e-2, which reflect the strength of the IB constraint.  While β=1e−7 corresponds to a weak IB constraint, it still provides a non-negligible regularization effect. Thanks again for your comment, which made us realize the necessity of including an additional ablation study that completely removes the IB constraint, *i.e.*, β=0. Herein, we provide comparison results between our full version and the ablated version as shown below.
>
> |bpp/PSNR|Rain_L|Haze|Noise_25|
> |-|-|-|-|
> |Ours|0.235/28.01|0.191/26.53|0.289/27.51|
> |Ablated|0.232/27.61|0.200/26.00|0.289/26.99|
>
> As shown, completely discarding the IB constraint leads to a notable performance drop. We will incorporate the complete results into the ablation studies section of our revised manuscript.
>
> Moreover, we would like to emphasize that our probabilistic prompt design is tightly coupled with the IB constraint. Specifically, the IB constraint requires estimating mutual information, and our probabilistic prompt represents degradations as a multivariate Gaussian distribution, which enables variational estimation of mutual information via the KL divergence.

---

### Official Review · Reviewer_oQak · 2025-06-29

**Clarity:** 3
**Significance:** 2
**Originality:** 1
**Rating:** 4
**Confidence:** 4

**Summary:**

This paper propose a lightweight plug-and-play module for degradation image compression. The author propose two module, Probabilistic Prompt Generator (PPG) and Degradation Adaptive Gating Modulation (DAGM) to align the latent distribution across degradation types and use Information Bottleneck principle for optimization.

**Questions:**

- In Fig.4, the results of restoration-then-compression paradigm use the same frozen LIC backbone, which made no adaptation for restored images. If using the restored image output by the restoration methods and the clean image pair to finetune the LIC model, is it can achieve better results, even better than the proposed methods? As the plug-and-play model needs to train for 100 epochs, a more convincing way is to compare with the adaptive LIC models which are finetuned using restored images.
- In line 275, “While both are capable of achieving satisfactory perceptual quality, they tend to result in a significant increase in bpp.” However, it seems the proposed method have higher bpps in low bpp regions. It’s better to explain why.
- Perceptual quality is usually measured by LPIPS or FID. The author use PSNR as metric, but use perceptual quality as description is confused.

**Ethical Concerns:**

["NO or VERY MINOR ethics concerns only"]

**Final Justification:**

Considering the technical reliability and practical performance, I raise my score to 4 Borderline accept.

**Limitations:**

See Questions section.

**Quality:**

2

**Strengths And Weaknesses:**

### Strength

- The proposed framework is unified for different degradation images.
- The plug-and play module can be used for different LIC backbones.

### Weakness

- The proposed module, PPG and DAGM seems to have little difference with PromptIR, which is lack of novelty.
- The results are worse than MOCE-IR in some cases.

---

> ### Author Rebuttal · Authors · 2025-07-31
>
> Q1: Concern about the novelty of PPG and DAGM compared to PromptIR
>
> Ans.: Thanks for your comments. We realize that your concern may stem from the lack of a direct comparison between our prompt learning strategy and PromptIR. To demonstrate the benefit of our design, we replaced our PPG module with the PromptIR and retrained it under the R-D criterion using identical hyperparameters for the LIC backbone (TIC). The comparsion result at the same compression level and the prompt sizes are presented below, where our approach shows a clear overall improvement.
> |PSNR|Rain_L(bpp=0.235)|Haze(bpp=0.191)|Noise_50(bpp=0.261)|Para.(M)|
> |-|-|-|-|-|
> |Probabilistic (Ours)|28.01|26.53|25.93|4.6|
> |Factorized|27.95|26.39|25.76|16.0|
>
> Moreover, as noted in our manuscript, PromptIR is the most closely related work to ours. Our key contribution is to further improve upon it in prompt design, learning strategy, and information theoretic foundation.
>
> To address your concern, we highlight our differences and enhancements over PromptIR in the following aspects: 1) *Factorized vs. probabilistic design*: PromptIR employs a set of factorized vectors to capture the characteristics of different degradation types. In contrast, our VAE style prompt generator models each degradation as the parameters of a unified multivariate Gaussian distribution. This fundamental shift lets us represent degradation characteristics with a much smaller prompt size, as shown in the table above. 2) *Training strategy*: PromptIR applies no specific constraints or supervision during prompt learning, and its naive prompt design hinders the learning of clean, discriminative degradation representations. By contrast, we employ the information bottleneck principle to impose a principled constraint on prompt learning, which enhances the prompt’s capacity to capture degradation characteristics. An intuitive comparison is shown in Fig. 1(e) and (f). 3) *Information-theoretic foundation*: Unlike PromptIR, which relies on empirical heuristics, our prompt is grounded in the information bottleneck principle. We design and optimize the probabilistic prompt according to a clear theoretical criterion: it encodes a minimal sufficient statistic of the degradation signal given the image content. To realize this design in practice, we investigate mutual information as a mathematical tool and derive variational bounds for efficient implementation.
>
> Q2. The results are worse than MOCE-IR in some cases.
>
> Ans.: Thanks for your comment. Indeed, MOCE-IR is a powerful “many-in-one” image restorer and delivers outstanding restoration results. However, in five of six tested degradation scenarios, our joint framework still outperforms the MOCE-IR. Moreover, as shown in Table 1 of our manuscript, our lightweight plug in offers overwhelming advantages in parameter scale (ours: 4.6 M *vs.* MOCE-IR:25.35 M) and inference speed (ours: 109.5 ms *vs.* MOCE-IR: 483.3 ms), which is critical for real world deployment.
>
> Q3. Concern about whether fine-tuning the LIC model on restored images would outperform the proposed method.
>
> Ans.: Thanks for your suggestions. It is true that fine tuning the LIC backbone on restored images can boost performance in certain test scenarios. However, this approach raises two key concerns. First, as noted in our manuscript, restored images exhibit different statistics from natural images. Fine tuning on them can therefore degrade compression performance on pristine inputs. Second, the fine tuning dataset is usually much smaller than the original LIC training set, which risks overfitting and poorer generalization to other scenarios.
>
> To illustrate these effects, we fine tuned the TIC backbone for 100 epochs using images restored by PromptIR. We then evaluated this model on (1) the Rain_H set from our manuscript and (2) a pristine image dataset RAISE1K (*RAISE: a raw images dataset for digital image forensics*) to assess its performance on normal real world inputs.
> |bpp/PSNR|Rain_L|Raise1K|
> |-|-|-|
> |Ours|0.235/28.01|0.187/31.61|
> |Finetuned|0.223/28.39|0.185/30.42|
>
> As shown, fine tuning the LIC backbone yields only marginal gains on our test set while inevitably causing overfitting and degrading compression performance on normal images.
>
> Moreover, in real world deployment scenarios, the training loss hyperparameters or even the training script may not be accessible, making the fine tuning strategy impractical.
>
> Q4. Concern about the higher bpp of the proposed method in low bitrate regions.
>
> Ans.: According to our analysis, in very low bitrate regions the LIC backbone must discard much of the high frequency content to meet the rate target, which leads to overly smooth reconstructions. In contrast, our method gradually adapts the compressed features to align with the pristine image distribution, restoring richer high frequency details. Encoding these additional restoration cues inevitably requires extra bits, which explains the modest bpp increase at the lowest rates. Thanks again for your comprehensive review in further improving the quality of our manuscript.  We will revise corresponding statement and include this explanation around line 275 in the updated manuscript to improve its accuracy.
>
> Q5. Incorporate more comparison results regarding perception-oriented metrics
>
> Ans.: Thanks for your valuable suggestion. Accordingly, we have provided partial results comparing our method with other baselines regarding LPIPS and FID on haze test set, as shown below.
>
> |LPIPS|bpp=0.143|bpp =0.191|bpp =0.228|bpp=0.243|
> |-|-|-|-|-|
> Ours(TIC)|0.1047|0.0910|0.0856|0.0830|
> |PromptIR+TIC|0.1112|0.1016|0.0954|0.0941|
> |AirNet+TIC|0.1283|0.1016|0.0987|0.0977|
> |MOCE-IR+TIC|0.1230|0.0952|0.0901|0.0867|
>
> |FID|bpp=0.143|bpp=0.191|bpp=0.228|bpp=0.243|
> |-|-|-|-|-|
> |Ours(TIC)|76.98|65.54|62.76|60.10|
> |PromptIR+TIC|92.49|68.97|65.14|60.00|
> |AirNet+TIC|90.32|72.47|65.12|63.16|
> |MOCE-IR+TIC|88.74|70.36|64.30|61.28|
>
> As shown in above table, Regarding both LPIPS and FID, our method also shown an constant advantages against the employed anchors.
> Since we cannot include full rate-perception curves in the rebuttal, we report only the results on *haze* test set here. The comprehensive comparison results would be provided in the supplementary materials of our revised manuscript. We appreciate your understanding.

---

> > ### Author Response · Authors · 2025-08-07
> > **Response to oQak**
> >
> > First of all, we are glad and encouraged to hear that our response addressed part of your concerns. We hope that our following responses adequately address your remaining concerns, and we welcome any further questions or feedback.
> >
> > *Q1. Concerns regarding the performance comparisons with PromptIR.*
> >
> > Ans.: Thanks for your kind comments. The IB constraint, as highlighted in our manuscript, regulates the mutual information between the learned prompt and input images, guiding the prompt to focus on degradation characteristics while minimizing interference from image content. This leads to **observable gains in in-domain comparisons and, more importantly, enhances generalization significantly** by promoting the extraction of essential features. Consequently, our approach outperforms PromptIR, which uses an empirically factorized prompt design, in unseen scenarios.
> >
> > To demonstrate this, we further include two out-of-domain comparisons:  1) **Mixed degradations (Haze + Rain):** It is usually seen in real-world scenarios. Specifically, we use the haze test set from our manuscript and add rain degradations with the method from *"Joint Rain Detection and Removal from a Single Image with Contextualized Deep Networks"* (TPAMI 2019). 2) **Unseen domain:** We also adopt an unseen deraining dataset, DID *(“Density-aware Single Image De-raining using a Multi-stream Dense Network”)*, and evaluate on its test set (1200 pristine–rainy image pairs).
> >
> > The corresponding results are listed in the table below.
> >
> > |bpp/PSNR | Rain+Haze | Unseen Domain (DID) |
> > | :--- | :--- | :--- |
> > | Ours | 0.183/24.09 | 0.202/25.15 |
> > | PromptIR | 0.207/19.24 | 0.214/22.93 |
> >
> > Our method shows consistent improvements, highlighting strong generalization for real-world deployment.
> >
> > Thank you for your insightful comments, which have greatly improved our manuscript. We will include the PromptIR comparison results in the supplementary materials to highlight our design's effectiveness.
> >
> >  *Q2. It’s not reasonable to use a finetuned model of degradation images to compress the nature images. And the experiment shows that finetune can truely get a better result.*
> >
> > Ans.: Thanks for your kind comments. First, we would like to emphasize that compressing natural images is a core encoder function, which also motivates our IB method. In real-world use, codecs primarily handle natural images, so any adaptation scheme must maintain high performance on them.
> >
> > As demonstrated in our previous response, fine-tuning the Learned Image Compression (LIC) backbone on restored images can lead to overfitting to specific scenarios, which degrades performance on natural images. Thanks to the excellent properties of IB to extract intrinsic features, our experiments show that our method can automatically identify non-degraded inputs based on the distribution parameters captured by the PPG module, thereby preserving performance on normal scenes with less compromise.
> >
> > Finally, encouraged by your feedback, we conducted an additional experiment to further investigate this under a more fair comparison setting. Specifically, we first feed pristine images into the restoration model, followed by compression using the fine-tuned LIC backbone. The results are presented in the table below also clearly demonstrate our superiority.
> >
> > | bpp/PSNR | Rain_L | Raise1K(Pristine) | Raise1K(Restored) |
> > | :--- | :--- | :--- | :--- |
> > | Ours | 0.235/28.01 | 0.187/31.61 | - |
> > | Finetuned| 0.223/28.39 | 0.185/30.42 | 0.184/28.26 |
> >
> > *Q3. What’s more, the method proposed in this paper also need training loss hyperparameters and the training script to train the PPG and DALM module.*
> >
> > Ans.: Thanks for your comments. First, we would like to clarify that our proposed method does not rely on the training script of the LIC backbone, as our plug-and-play design does not involve modifying any of its parameters. As for the training hyperparameters, they can be empirically selected based on final performance.
> >
> > Beyond this point, we would like to summarize and highlight the superiority of our method over the fine-tuning approach as follows: 1) compatibility with normal scenes without compromising performance, and 2) lower computational cost and inference time compared to fine-tuning strategies, as detailed in Table 1 of our manuscript..

---

> > > ### Comment · Reviewer_oQak · 2025-08-08
> > >
> > > Thank you for the clarification and additional experiments, especially the mixed degradation and unseen domain scenarios, which shows strong generalization performance.
> > > Considering the technical reliability and practical performance, I raise my score to 4  Borderline accept.

---

> ### Comment · Reviewer_oQak · 2025-08-06
>
> Thank you for the authors to provide answers and additional experimental results.
>
> Positive revisions:
>
> - Explanation on higher bpp of the proposed method in low bitrate regions. I acknowledge that aligning the distribution with the pristine image needs high frequency details which may lead to higher bbpp.
> - Experiments considering LIPIPS and FID show great result compared with restoration methods combine with compression methods.
>
> Remaining concerns:
>
> - The results indicate that replacing PPG module with the PromptIR does not lead to a serious degradation of image quality. At the same compression level, the PSNR drops by about 0.1. I understand that the authors have done some training strategy based on theoretic and improve the PromptIR prompt design, but it only seems reduce the prompt parameters with no obviously image quality increase. However, both 4.6M and 16M are relatively small in scale.
> - It’s not reasonable to use a finetuned model of degradation images to compress the nature images. And the experiment shows that finetune can truely get a better result. What’s more, the method proposed in this paper also need training loss hyperparameters and the training script to train the PPG and DALM module.

---

### Official Review · Reviewer_d5Gs · 2025-07-01

**Clarity:** 2
**Significance:** 2
**Originality:** 2
**Rating:** 2
**Confidence:** 5

**Summary:**

The authors propose a prompt tuning method to finetune an image codec, enabling it to simultaneously perform image restoration tasks (such as denoising, deraining, dehazing, etc.) and image compression.

**Questions:**

Please answer the questions in the above weaknesses.

**Ethical Concerns:**

["NO or VERY MINOR ethics concerns only"]

**Final Justification:**

Thank you for the authors' response. However, I remain unconvinced.

1) I still hold my original view: rain and fog are natural phenomena and should be preserved as valid information—even in surveillance scenarios. Regarding the authors' claim that such degradations lead to unstable bitrate, the proper solution is not to remove rain or fog, but to design image encoders that maintain stable bitrate even under such conditions (while preserving rain and fog in the reconstructed image). As for machine analysis tasks, the authors did not include any relevant experiments in the main paper. Instead, all comparisons are based on PSNR-measured rate-distortion performance. This work is not a "image coding for machine" approach. The few additional experiments added in the rebuttal lack sufficient depth and credibility, and they are not well integrated into the existing narrative of the paper.
2) Although the authors highlight some differences from prior works [a, b, c], these differences do not represent fundamental improvements. Therefore, the novelty of this work remains insufficient.
3) While the authors have supplemented a few comparison experiments with joint denoising-and-compression methods in the rebuttal, the overall experimental setup in the paper still primarily compares with the unfair "restoration-then-compression" paradigm. Performing substantial revisions would go beyond the scope allowed for conference papers.
4) The authors have added some comparisons with other fine-tuning strategies, but these experiments are still too limited in scope and depth.


In summary, the motivation and novelty of this work are unconvincing, and the experimental setup is not fair. Although the authors have attempted to address concerns in the rebuttal, the supplementary data are insufficient, and making substantial revisions to the paper would not be appropriate under the constraints of a conference submission. I maintain my decision to reject the paper and recommend that the authors thoroughly revise it for submission to another conference.

**Limitations:**

The authors do not discuss the limitations and potential negative societal impact of their work.

**Paper Formatting Concerns:**

None.

**Quality:**

2

**Strengths And Weaknesses:**

Strengths: The authors propose a prompt tuning method to finetune an image codec to achieve joint image restoration and compression.

Weaknesses:
1) Is it necessary to combine image restoration tasks, such as deraining and dehazing, with image compression? Rain and haze are common natural phenomena and may well be part of the information the user intends to record. The goal of image compression is to reduce storage bitrate while preserving image fidelity as much as possible. Enforcing deraining or dehazing during the compression process may go against the user's intention and destroy the value of the original image.
2) Using prompting tuning to finetune image compression for domain and task adaption have been studied in previous works, such as [a,b,c]. The novelty of this work is limited.
3) Joint image denoising and compression is reasonable and has been studied in previous works, such as [d,e]. The authors should compare with these related methods. It is unfair to compare with restoration-then-compression scheme.
4) What would happen if the authors' PPG and DALM were replaced with fine-tuning strategies such as AdaIN or LoRA? Is the prompt tuning strategy proposed by the authors the best possible approach?

[a] Yi-Hsin et al., TransTIC: Transferring Transformer-based Image Compression from Human Perception to Machine Perception, ICCV, 2023.

[b] Han Li et al, Image Compression for Machine and Human Vision with Spatial-Frequency Adaptation, ECCV, 2024.

[c] Koki Tsubota et al., Universal Deep Image Compression via Content-Adaptive Optimization with Adapters, WACV, 2023.

[d] Benoit Brummer, Christophe De Vleeschouwer, On the Importance of Denoising When Learning to Compress Images, WACV, 2023.

[e] Xi Zhang, Xiaolin Wu, FLLIC: Functionally Lossless Image Compression, arxiv, 2024.

---

> ### Author Rebuttal · Authors · 2025-07-31
>
> Q1: Concerns reading the necessity of combining image restoration with image compression
>
> Ans.: Thanks for your comments. In response to your concern, we’d like to summarize the motivation and necessity of developing the joint restoration-compression framework as following:
>
> 1)**Maintain stable bitrates for degraded scenarios:** LIC models are trained on pristine images. Compressing degraded inputs such as rainy or noisy scenes therefore often requires a significantly higher bitrate as their statistics differ substantially from those of the training data (as shown in Fig. 1 of our manuscript).  In real time transmission scenarios, such as surveillance, maintaining a stable bitstream is crucial. Fluctuations caused by environmental degradations can lead to network congestion, packet loss, and even the loss of critical frames. By integrating lightweight restoration into compression, our method keeps the bitrate within the expected range.
>
> 2)**Necessity for machine vision-oriented scenarios:** Rain and haze severely degrade the accuracy of downstream vision tasks. To demonstrate this, we evaluated Fast R-CNN on real-world traffic scene object detection using the DAWN dataset (*DAWN: Vehicle Detection in Adverse Weather Nature*). Specifically, we selected subsets containing fog (300 images) and rain (200 images) degradations for evaluation.
>
> |Rain|bpp|mAP50|
> |-|-|-|
> |Ours (TIC)|0.177|35.25%|
> |PromptIR+TIC|0.182|30.33%|
> |TIC|0.209|28.11%|
>
> |Fog|bpp|mAP50|
> |-|-|-|
> |Ours (TIC)|0.116|33.62%|
> |PromptIR+TIC|0.124|25.23%|
> |TIC|0.100|21.92%|
>
> As shown in above table, our joint framework achieves a better rate–analytics performance than both vanilla LIC and a sequential restoration then compression pipeline. Moreover, our joint strategy with its lightweight design offers clear coding latency advantages over the two stage pipeline (see Table 1 of our manuscript), which is crucial for real world deployment.
>
> 3)**User controlled flexibility:** Moreover, we acknowledge that in some applications users may wish to preserve rain or haze as part of the scene content. Owing to our plug and play design, users who prefer a fidelity-driven compression process can disable the restoration module and employ the original LIC model without modification.
>
> Q2. Concerns about the novelty over existing prompt-tuning methods for domain and task adaptation in image compression
>
> Ans. Thanks for your recommendation of all of these brilliant works, that all involves in deploying plug-and-play modules at the LIC backbones for domain or task adaptation. After a comprehensive study of their papers and open-sourced code, we believe our work demonstrates significant differences and novelty in terms of technical design, theoretical foundation, and implementation flexibility. Specifically,
>
> 1)**Task/domain specific *vs*. generalized:** Our IB-LRU scheme is especially designed to handles multiple degradation types in a single training process. By contrast, methods in [a] and [b] insert task-specific modules that must be trained separately for each target, after a comprehensive study of their open-source code. In particular, although these methods demonstrate effectiveness across multiple tasks, each downstream task requires its **own prompt**, a **separate compression pass**, and an **individual bitstream**. Meanwhile, [c] employs an online optimization strategy that uses gradient information from the decoder to refine the latent representation during each image’s test time encoding. This incurs unpredictable latency and requires the encoder and decoder to be co-located, which is impractical for many deployments. For intuitive demonstration, we evaluate its adaptation capacity using the H_Rain test set. Specifically, we use its open-source checkpoint to compress paired pristine and rainy images. The resulting bpp, fidelity performance, and inference speed (s) are listed below.
>
> |bpp(Clean)|bpp(Rain)|PSNR(Clean)|PSNR(Rain)|Inf.speed ([c])|Inf. speed (Ours)|
> |-|-|-|-|-|-|
> |0.274|0.521|29.83|27.41|283s|0.103s|
>
> As shown, although [c] incorporates an online optimization strategy at the cost of significantly increased coding latency, it still struggles with open-world degradations, resulting in a substantial increase in bpp and a noticeable loss in fidelity.
>
> 2)**Encoder only design:** Works [a], [b], and [c] modify both encoder and decoder. Meanwhile, we only alter the encoder, leaving the decoder unchanged. This design allows our module to be dropped into existing LIC systems without any changes on the client or server side, which is critical for real world adoption.
>
> 3)**Information Theoretic Prompt Design:** We emphasize our theoretical contributions in designing the probabilistic prompt and the IB constraint. Unlike existing works, which rely on empirical heuristics, our prompt is grounded in the information bottleneck principle: it encodes a minimal sufficient statistic of the degradation given the image content, then we devote plenty of work to derive variational bounds to realize this design in practice.
> To illustrate the benefit of our design, we straightforwardly replaced our PPG module with the commonly employed factorized prompt (PromptIR) and retrained it using the R-D criterion. As shown in below table, our probabilistic prompt with the IB constraint outperforms the factorized design in both perceptual quality and computational complexity.
>
> |PSNR|Rain_L(bpp=0.235)|Haze(bpp=0.191)|Noise_50(bpp=0.261)|Para.(M)|
> |-|-|-|-|-|
> |Probabilistic(Ours)|28.01|26.53|25.93|4.6|
> |Factorized|27.95|26.39|25.76|16.0|
>
> Q3. Comparing with other joint denoising and compression works.
>
> Ans.: Thanks for your suggestion. As [e]’s source code has not been open sourced, we provide comparison results with [d]. Specifically, since [d] focuses solely on denoising, we retrained both our scheme and [d] (*JDC-CN* version) using the same LIC backbone TIC and the same training set (containing only Gaussian noise). The comparison results regarding the bpp and PSNR (dB)  on the Gaussian noise test set are shown below.
>
> |bpp/PSNR|Noise_15|Noise_25|Noise_50|
> |-|-|-|-|
> Ours|0.271/28.23|0.294/28.07|0.245/26.53|
> [d]|0.254/24.71|0.283/24.20|0.265/23.01|
>
> As shown, even under single-noise settings, our method still demonstrates notable advantages. This is expected, as [d] primarily focuses on a new training strategy and supervision terms while building upon existing LIC models. While it makes a valuable contribution, its primary goal is methodological insight rather than pushing performance to the SOTA.
>
> Q4. Comparing with other fine-tuning strategies.
>
> Ans.: Thanks your suggestion. Due to time and computational resource limitations, we were only able to implement the most recent method, LoRA. For a fair comparison, we applied LoRA (*lora_rank*=16) only on the encoder side of the same LIC backbone checkpoint as ours (TIC), and trained the LoRA module on the same dataset used in our method. The testing results regrading bpp and PSNR (dB) are shown below,
>
> |bpp/PSNR|Haze|Rain_L|Noise_50|
> |-|-|-|-|
> Lora(TIC)|0.175/24.12|0.238/26.47|0.283/23.89|
> |Ours(TIC)|0.191/26.53|0.235/28.01|0.261/25.93|
>
> As shown, although LoRA achieves strong performance when adapting to a specific task or domain, it is not designed to generalize across multiple degradation types within a single training process. In contrast, our PPG and IB constraint are specifically designed for multi-degradation scenarios, offering notable improvements and novelty beyond existing works, while also being supported by a solid information-theoretic foundation.

---

> > ### Comment · Reviewer_d5Gs · 2025-08-08
> >
> > Thank you for the authors' response. However, I remain unconvinced.
> >
> > 1) I still hold my original view: rain and fog are natural phenomena and should be preserved as valid information—even in surveillance scenarios. Regarding the authors' claim that such degradations lead to unstable bitrate, the proper solution is not to remove rain or fog, but to design image encoders that maintain stable bitrate even under such conditions (while preserving rain and fog in the reconstructed image). As for machine analysis tasks, the authors did not include any relevant experiments in the main paper. Instead, all comparisons are based on PSNR-measured rate-distortion performance. This work is not a "image coding for machine" approach. The few additional experiments added in the rebuttal lack sufficient depth and credibility, and they are not well integrated into the existing narrative of the paper.
> > 2) Although the authors highlight some differences from prior works [a, b, c], these differences do not represent fundamental improvements. Therefore, the novelty of this work remains insufficient.
> > 3) While the authors have supplemented a few comparison experiments with joint denoising-and-compression methods in the rebuttal, the overall experimental setup in the paper still primarily compares with the unfair "restoration-then-compression" paradigm. Performing substantial revisions would go beyond the scope allowed for conference papers.
> > 4) The authors have added some comparisons with other fine-tuning strategies, but these experiments are still too limited in scope and depth.
> >
> >
> > In summary, the motivation and novelty of this work are unconvincing, and the experimental setup is not fair. Although the authors have attempted to address concerns in the rebuttal, the supplementary data are insufficient, and making substantial revisions to the paper would not be appropriate under the constraints of a conference submission. I maintain my decision to reject the paper and recommend that the authors thoroughly revise it for submission to another conference.

---

### Official Review · Reviewer_Tuoa · 2025-07-03

**Clarity:** 3
**Significance:** 3
**Originality:** 3
**Rating:** 4
**Confidence:** 3

**Summary:**

This paper addresses the performance degradation of LIC models under image degradations, including noise and blur. Instead of using a separate restoration module, the authors propose learning degradation-aware prompts that adaptively condition the compression process. These latent conditioning signals guide the encoder to handle diverse degradations better. Additionally, an information bottleneck constraint ensures that the prompts capture essential degradation features while minimizing redundant content information.

**Questions:**

- How sensitive is the model's performance to the choice of the latent prompt dimensionality d? Can authors provide the experiments with different prompt sizes?

- Do authors have any insights into how the IB constraint weight should be tuned for unseen degradation types?

**Ethical Concerns:**

["NO or VERY MINOR ethics concerns only"]

**Final Justification:**

Most of my concerns have been addressed in the rebuttal, and I appreciate the authors’ clarifications. However, after reading the exchange between the authors and other reviewers, I noticed one aspect that could still be improved.

The discussion around integrating restoration (e.g., deraining, dehazing) into compression could benefit from a more user-centric perspective. As reviewer d5Gs pointed out, degradations like rain or haze may be part of the scene that users intentionally capture. Automatically removing them during compression may not always align with user intent. While the authors justify this approach as a means to maintain a stable bitrate, a more desirable research direction could be to preserve the original content while still achieving efficient and stable compression.

That said, the paper is practically motivated and technically sound overall. I therefore maintain my original score of 4: Borderline Accept.

**Limitations:**

No. Since the checklist guides the authors to include a separate Limitations” section, the authors are recommended to add one in the paper. Potential limitations may include the dependence on frozen LIC backbones and the limited diversity of degradation types evaluated.

**Paper Formatting Concerns:**

No major formatting issues noticed.

**Quality:**

3

**Strengths And Weaknesses:**

- **Strength**

  - **Prompt-based latent representation alignment**: The paper introduces a new approach that generates probabilistic prompts to modulate encoder features adaptively, enabling better alignment between degraded and clean latent distributions.

  - **Integration of the information bottleneck principle**: The authors apply the IB principle to control the mutual information in prompt learning, ensuring that the prompts capture essential degradation signals while suppressing redundant content.

  - **Effective bitrate reduction**: The proposed IB-LRU scheme demonstrates substantial bitrate reductions and perceptual quality improvements across multiple degradation types.

   - **Lightweight and flexible design**: Unlike cascaded restoration-then-compression pipelines, the method requires fewer additional parameters and achieves significantly faster inference.


- **Weakness**

  - **Limited diversity of degradation types**: Although the paper covers haze, rain, and noise, it does not evaluate more complex or domain-specific degradations (e.g., heavy compression artifacts, under-display camera artifacts, or motion blur), which may limit the generalizability of the approach.

  - **Dependence on frozen LIC backbones**: The method assumes pre-trained LIC models remain fixed during training, which could restrict potential improvements that might arise from jointly fine-tuning the backbone along with the prompt module.

---

> ### Author Rebuttal · Authors · 2025-07-31
>
> First of all, we would like to express our sincere appreciation for your positive attitude and constructive suggestions on our work. Your feedback is highly valuable to us and serves as a strong source of encouragement. We hope the following point-by-point responses successfully address your concerns, and we look forward to any further suggestions you may have.
>
> Q1: Limited diversity of degradation types
>
> Ans.: Thanks again for your constructive suggestions. Accordingly, we first evaluate our method under a complex degradation scenario of combined haze and rain, which frequently occurs in real world applications. Specifically, we use the haze test set from our manuscript and add rain degradations using the approach from *"Joint Rain Detection and Removal from a Single Image with Contextualized Deep Networks"* (TPAMI 2019). Comparison results between ours and the employed restoration-then-compression anchor are shown below.
>
> ||bpp/PSNR|
> |-|-|
> |Ours (TIC)|0.183/24.09|
> |Prompt+TIC|0.187/18.90|
>
> These results clearly demonstrate the effectiveness of our method in complex scenarios.
>
> Moreover, to evaluate a different degradation type, we investigate the common low light condition. We incorporated the training set of LOLv2 *(Low-light image enhancement via structure modeling and guidance)* and LOD *(Crafting Object Detection in Very Low Light )* into our training stage and retrained our method. The testing results regarding bpp and PSNR(dB) are listed below.
>
> ||bpp/PSNR|
> |-|-|
> |Ours (TIC)|0.138/20.33|
> |Prompt+TIC|0.132/18.71|
>
> Due to time and computational resource limitations, we will provide additional results on other domain specific degradations in our supplementary materials. We appreciate your understanding.
>
> Q2: Dependence on frozen LIC backbones
>
> Ans.: Thanks for your kind suggestions. To comprehensively examine the effectiveness of relaxing the parameters of the backbone network, we incorporate two additional comparisons: a）Jointly training: We release the LIC backbone from the beginning and jointly train it with our proposed modules, keeping all other settings exactly the same. b) Finetuning, based on our current version, we released the parameters of the TIC backbone and fine tuned the entire framework for an additional 100 epochs. The comparison results on the compression level of bpp=0.191 are listed below.
>
> |PSNR(dB)|Haze|COCO|
> |-|-|-|
> |Ours|26.53|28.96|
> |Jointly|24.95|25.88|
> Finetuning|26.90|27.26|
>
> First, regarding the joint training scheme, it even leads to reduced performance. According to our analysis, this is mainly because jointly training from scratch disrupts the prior knowledge of pristine images contained in the LIC backbone, which in turn prevents our prompt from effectively decoupling the degradation information.
>
> Moreover, finetuning the backbone can indeed improve performance on certain test cases. However, we would like to emphasize our motivation for adopting the plug-and-play strategy, as well as its necessity in practical deployment scenarios. 1) First, as noted in our manuscript, we take degraded images as input, which exhibit significantly different statistics from natural images. As a result, fine tuning on such inputs can degrade compression performance on pristine images. This outcome is unacceptable in real world scenarios, where most content consists of normal, undegraded scenes. In contrast, our probabilistic prompt design can effectively recognize undegraded inputs through the learned distribution parameters and adapt accordingly. 2) Second, the fine tuning dataset is typically much smaller than the original LIC training set, which increases the risk of overfitting and reduces generalization to other conditions.
>
> To further validate this, we evaluated the performance on an additional pristine dataset COCO. The comparison results are listed in the table above. As shown, the fine-tuning strategy inevitably leads to a performance drop on normal scenes. Given these issues, recent works across a wide range of research domains increasingly favor plug and play approaches over backbone fine tuning.
>
> Q3 Discussion on the impact of different prompt dimensionalities (d) on performance
>
> Ans.: Thank you for your constructive suggestions, which are indeed necessary and have helped improve the quality of our manuscript. Since our proposed probabilistic prompt is designed to capture discriminative characteristics of different degradations, we chose the prompt size based on a more straightforward and intuitive evaluation using t-SNE visualizations, rather than solely focusing on final compression performance (An example of t-SNE plot is shown in Fig. 1 (f) of our manuscript). We tested a range of prompt sizes *d*=64, 128, 256, 512 and selected *d*=128 as it already yielded satisfactory representation. Increasing the size further brought minimal improvement while significantly increasing the model’s parameter scale. For intuitive demonstration, we list the parameter scale corresponding to different choices of d below.
>
> |*d*|64|128|256|512|
> |-|-|-|-|-|
> Par.#|4.2M|4.6M|6.2M|12.0M|
>
> Due to the rebuttal constraint, we are not capable of improve the visualize result but we will incorporate to our revised manuscript as an ablation study, thanks again for your suggestions.
>
> Q4.: Guidance on tuning the IB constraint weight for unseen degradations
>
> Ans.: Thanks a lot for your kind suggestions. First, we would like to clarify that our scheme relies on the training process to capture the distribution of different degradation types. From a theoretical standpoint, our method is not capable of being directly applied to degradation scenarios whose patterns differ significantly from those in the training set. However, our lightweight design offers strong flexibility and can be easily adapted to unseen scenarios by expanding the training set, with limited computational overhead. As shown in the experimental results provided in response to your first question, our method achieves satisfactory performance in low-light scenes after only 20 epochs of training (less than 2 GPU Hours). In the context of incorporating new degradation types, the IB constraint needs to be relaxed. Specifically, by using a smaller β value, since the prompt is expected to capture more information to represent the added complexity.
>
> Moreover, in simpler cases, such as complex degradations composed of combinations of seen degradation types, or unseen intensities within known types, our method can be directly applied without retraining. This is demonstrated in the haze+rain scenario shown in the experimental results provided in response to your first question, as well as in the test results on the RAIN test set in our manuscript, where the rain degradation intensity spans a wide range. In this context, there is no need to adjust the IB constraint.

---

> ### Comment · Reviewer_Tuoa · 2025-08-06
>
> Most of my concerns have been addressed in the rebuttal, and I appreciate the authors’ clarifications. However, after reading the exchange between the authors and other reviewers, I noticed one aspect that could still be improved.
>
> The discussion around integrating restoration (e.g., deraining, dehazing) into compression could benefit from a more user-centric perspective. As reviewer d5Gs pointed out, degradations like rain or haze may be part of the scene that users intentionally capture. Automatically removing them during compression may not always align with user intent. While the authors justify this approach as a means to maintain a stable bitrate, a more desirable research direction could be to preserve the original content while still achieving efficient and stable compression.
>
> That said, the paper is practically motivated and technically sound overall. I therefore maintain my original score of 4: Borderline Accept.

---

### Decision · Program_Chairs · 2025-09-17

**Decision:**

Accept (poster)

**Comment:**

This paper received mixed initial ratings (1 accept, 2 borderline accepts, 1 reject). There was general appreciation for the problem motivation, technical soundness and overall performance and versatility of the proposed approach in achieving bitrate reduction.

There were several concerns raised -- including potential lack of generalization and comparisons with existing methods --- most of which were addressed by the authors.

There remained a lingering concern around the proposed method "forcibly" removing potential sources of image degradation (e.g., rain). The authors did clarify though that their approach has a flexible option to perform deraining or to keep these cues, depending on task requirements.

There was significant discussion between the authors and reviewers, and also among the reviewers and the AC post rebuttal. While a consensus was not reached, after considering all the points, on balance, an accept decision was reached.

The authors are urged to carefully consider the reviewer comments (especially those from Reviewer d5Gs) when preparing the camera-ready version.